# Contactin 1: An Important and Emerging Oncogenic Protein Promoting Cancer Progression and Metastasis

**DOI:** 10.3390/genes11080874

**Published:** 2020-07-31

**Authors:** Yan Gu, Taosha Li, Anil Kapoor, Pierre Major, Damu Tang

**Affiliations:** 1Department of Medicine, McMaster University, Hamilton, ON L8S 4K1, Canada; guy3@mcmaster.ca (Y.G.); akapoor@mcmaster.ca (A.K.); 2The Research Institute of St Joe’s Hamilton, St. Joseph’s Hospital, Hamilton, ON L8N 4A6, Canada; 3Urological Cancer Center for Research and Innovation (UCCRI), St. Joseph’s Hospital, Hamilton, ON L8N 4A6, Canada; 4Life-Tech Industry Alliance, Shenzhen 518000, China; litaosha@genomics.cn; 5Department of Surgery, McMaster University, Hamilton, ON L8S 4K1, Canada; 6Department of Oncology, McMaster University, Hamilton, ON L8S 4K1, Canada; majorp@hhsc.ca

**Keywords:** Contactin 1 (CNTN1), tumorigenesis, cancer progression, metastasis

## Abstract

Even with recent progress, cancer remains the second leading cause of death, outlining a need to widen the current understanding on oncogenic factors. Accumulating evidence from recent years suggest Contactin 1 (CNTN1)’s possession of multiple oncogenic activities in a variety of cancer types. CNTN1 is a cell adhesion molecule that is dysregulated in many human carcinomas and plays important roles in cancer progression and metastases. Abnormalities in CNTN1 expression associate with cancer progression and poor prognosis. Mechanistically, CNTN1 functions in various signaling pathways frequently altered in cancer, such as the vascular endothelial growth factor C (VEGFC)-VEGF receptor 3 (VEFGR3)/fms-related tyrosine kinase 4 (Flt4) axis, phosphatidylinositol 3-kinase (PI3K)/protein kinase B (AKT), Notch signaling pathway and epithelial-mesenchymal transition (EMT) process. These oncogenic events are resulted via interactions between tumor and stroma, which can be contributed by CNTN1, an adhesion protein. CNTN1 expression in breast cancer correlates with the expression of genes functioning in cancer-stroma interactions and skeletal system development. Evidence supports that CNTN1 promotes cancer-stromal interaction, resulting in activation of a complex network required for cancer progression and metastasis (bone metastasis for breast cancer). CNTN1 inhibitions has been proven to be effective in experimental models to reduce oncogenesis. In this paper, we will review CNTN1′s alterations in cancer, its main biochemical mechanisms and interactions with its relevant cancer pathways.

## 1. Introduction

Cell adhesion molecules (CAMs) are membrane receptors that bind to extracellular matrix molecules or receptors [1]. CAMs forms structural linkages and mediates interaction between the cell cytoskeleton and the extracellular matrix or between cells. This interaction facilitates the transduction of downstream signaling and regulate several important biological processes including cell division, migration and differentiation. The expression of CAMs are tightly regulated for refined control of cell proliferation, differentiation, mobility and survival in cellular processes. However, these processes may be hijacked and mis-regulated in tumorigenesis [1,2]. Aberrant expression of CAMs has been shown to associate with invasive phenotype of tumour cells. Metastatic dissemination is the leading cause of death for cancer patients worldwide [3]. This well-orchestrated process follows few key steps, including tumour cell invasion of the primary site, intravasation into blood and/or lymphatic vessels, dissemination into the circulation, and successful adherence and colonization to the final site of metastasis. For tumour cells to complete this metastatic cascade successfully, their ability for adaptation or plasticity needs to be enhanced or evolved, which is critically regulated through interactions with extracellular matrix (ECM) in the tumour microenvironment. Many CAMs play important roles in regulating these homotypic and heterotypic interactions [1,2,4].

Contactin 1 (CNTN1) is a neuronal cell adhesion molecule belonging to the subgroup of glycosyl phosphatidylinositol (GPI)-anchored neural cell adhesion molecules of the immunoglobulin superfamily (IgSF) [5]. Despite being primarily studied for its roles in the human nervous system, CNTN1 has been implicated in several signaling pathways altered in cancer. The aberrant activation of CNTN1 has been reported to be involved in pathological phenotypes such as cell proliferation, invasion, metastasis and poor prognosis [6,7]. On these premises, the interest of CNTN1 as a target to impede cancer progression is increasing. This review analyzes the up-to-date knowledge supporting CNTN1′s involvement in cancer progression and metastasis, and its related molecular mechanisms. In accordance with the PRISMA guidelines, we performed a systemic literature search through the PubMed database using the term “Contactin 1” and “cancer”. A total of 43 manuscripts from 2006 to 2020 were retrieved. We examined all abstracts and eliminated those: (1) studies yielding values of *p* > 0.05; (2) irrelevant to cancers and (3) retracted articles. We thus selected and discuss 33 articles in this review (Figure 1). The topic of CNTN1 in cancer progression has been previously reviewed [6,8], however some recent findings have been made and will be discussed.

## 2. The Role of CNTN1 in Neuronal Development

Of the six members of the CNTNs, Contactin 1 (CNTN1) was the first identified and the most thoroughly investigated for its function in pathological disorders [9]. CNTN1 is structurally comprised of 6 N-terminal Immunoglobulin (Ig) C2 domain, followed by 4 fibronectin type III (FNIII) repeats on the C-terminal, and a hydrophobic C-terminal amino acid sequence. This structure is common in GPI-linked proteins as it allows for lipid anchorage to the extracellular matrix (Figure 2) [9]. CNTN1 is mostly expressed in the human brain and neuronal tissues and is limitedly expressed in other tissues. CNTN1 primarily functions as an axonal glycoprotein important in the facilitation of axonal growth and the outgrowth of neurites [10,11], but also plays important roles in other neuronal developmental processes, such as differentiation and development of glial cells, myelination, synaptogenesis, and fasciculation [11,12,13,14]. The CNTN1 knockout transgenic mice shows a cerebellar phenotype in which the parallel fibers of the granule cell neurons are misoriented, concurrent with a 25% reduction on cerebellar volume, supporting its importance in axon guidance [15]. Specifically, the CNTN1 mutants show an overt ataxia phenotype by day 10 which subsequently progresses to pronounced smaller size, emaciated appearance and uncontrolled movement by day 15. The CNTN1 homozygous mutants were lethal by postnatal day 18, which may be attributable to impaired cerebellar functions, which interfere with normal feeding behavior.

CNTN1 binds with several extracellular matrix molecules (Figure 3). The association of CNTN1 with tenascin-C, tenascin-R, and receptor protein tyrosine phosphatase β (RPTPβ) promotes axonal growth and fasciculation which are pivotal in neuron-glial integration [16]. CNTN1 on the axon surface associates in *trans* with transmembrane protein tyrosine receptor type Z (PTPRZ) on the glial cells, in turn facilitates neurite outgrowth and glial adhesion [17,18,19,20]. CNTN1 also modulates the maturation and proliferation of oligodendrocyte precursor cells (OPCs) via its interaction with PTPRZ and Notch receptor [14,21,22]. When binding laterally in *cis* with the receptor protein tyrosine phosphatase α (RPTPα), CNTN1 promotes transduction of extracellular signals of tyrosine kinase FYN and influences cell mobility [23]. Interestingly, whist the interaction of CNTN1 with RPTPß substrate stimulates dorsal root ganglion (DRG) sensory axons extension [19,24], its association with Tenascin-R substrate actually inhibits cerebellar granule neuron receptor [25], suggesting the functions of CNTN1 may vary depending on cellular and molecular context. Recent study has identified an essential role of CNTN1 in radical neuronal migration and final localization of neurons in the developing mouse neocortex. The knockdown of CNTN1 via in utero electroporation resulted in significantly delayed radical migration mediated by RhoA activation, a Rho-GTPase critical for actin cytoskeletal reorganization [26]. Clinically, the expression of CNTN1 is dramatically reduced in patients with age-related memory decline [27], suggesting a role of CNTN1 in memory associated processes. A previous report also described a lethal form of congenital myopathy possibly caused by the loss of CNTN1 in the neuromuscular junction due to a familial mutation [28]. Collectively, these observations reveal that CNTN1 plays a critical role in the development of central nervous system through its adhesion functions. Adhesion is an essential process regulating tumorigenesis and cancer progression; in this regard, it is important to revisit the evidence supporting CNTN1-associated oncogenic functions.

## 3. Relevance of CNTN1 in Tumorigenesis

Neural cell adhesion molecules of the IgSF are complex membrane-anchored proteins that mediates multiple molecular interactions and include CNTNs, neural cell adhesion molecule (NCAM), L1, and neuron-glia related (Nr)-CAM [29]. It has become increasingly clear that changes in the adhesive properties of cancer cells, in terms of both cell–cell adhesion and adhesion to the extracellular matrix, are involved in tumor progression. Previous reports have supported that certain members of the Ig superfamily may be important in facilitating tumour invasion and metastasis [30,31]. The first CAM identified is NCAM, which contains 5 Ig-like and 2 fibronectin type III repeats with 2 major transmembrane isoforms and a GPI-linked isoform. Studies have shown NCAM to be commonly upregulated and promotes an adhesion switch during EMT that is associated with cancer invasion. Enforced expression of NCAM promotes mesenchymal-like properties, as evident by the reciprocal staining of E-cadherin in NCAM from RipTag2 mouse model of pancreatic cancer and aberrant persistence of E-cadherin expression resulted from NCAM deficiency. Furthermore, NCAM localized in membrane compartments binds and stimulates fibroblast growth factor receptor 1 (FGFR1) through its fibronectin type III (F3) domains and activates MAPK-mediated cell migration in non-neural cells. These findings imply a strong association of NCAM with EMT in human cancers. Indeed, the correlation between deregulated NCAM expression and poor prognosis has been found in different cancer types including neuroblastoma [32,33], myeloma [34,35], acute myeloid leukemia [36,37], pancreatic cancer [38,39,40], and small cell lung cancer [41].

Sharing structural and functional homology with NCAM, the role of CNTN1 outside of the nervous system was largely unexplored until recently. Northern blots analysis has shown low levels of CNTN1 transcripts in several tissues including pancreas, lung, kidney, and skeletal muscles but how CNTN1 functions in these tissues remains unexplored [42]. However, as the central nervous system was the only defective organ in CNTN1 deficient mice [15], the protein is not a major contributor to the development of other organs at least in mice. The CNTN1 gene locus is mapped on chromosome 12q11-q12 and is the only IgSF genes that is located at this position. Intriguingly, several other genes, which are mapped in positions between 12q11 and q13, have all been implicated in human cancers. These include the homeobox protein HOX3 [43], the integrin α5 subunit [44], collagen type II [45] and the proto-oncogenes *erb-b3* [46] and *int*-1 [47]. Intriguingly, integrin α5 subunit function in the cellular signaling emanated from adhesion [48], supporting the involvement of the CNTN1 genes residing in this region in cancer adhesion. Moreover, the chromosome position *12q13* is a heritable fragile site implicated in embryonic lethality [49] and is also a frequent breakpoint in several cancers [50]. Amplifications and gains of genomic materials are frequent in this region for malignant gliomas [51] and astrocytoma [52]. The location of CNTN1 gene locus close to a tumor breakpoint provides a potential genetic evidence for CNTN1 to function in tumor formation and progression. Moreover, CNTN1 was identified as a direct downstream molecule of the vascular endothelial growth factor C (VEGFC)-VEGF receptor 3 (VEFGR3)/fms-related tyrosine kinase 4 (Flt4) axis important in lymphangiogenesis and lymphatic invasion in cancers [53,54,55]. With growing interest in the investigation of CNTN1 for its cancer related functions, CNTN1 expression was found to be upregulated in many cancers including lung [56,57,58,59], gastric [60,61], breast [62], prostate [63,64], stomach [65], thyroid [7], esophageal [66] cancers, hepatocellular carcinoma [67,68], astrocytic gliomas [69], esophageal squamous cell carcinoma (ESCC) [70], and oral squamous cell carcinoma (OSCC) [71]. Unsurprisingly, increased CNTN1 expression was found to have functional associations in promoting cancer cell progression and metastasis. In these subsections below, we will discuss in details current studies that have investigated CNTN1′s role in human cancers (Table 1).

### 3.1. Lung Cancer

The role of CNTN1 as a metastasis-promoting oncogene was described by Su et al. in a genome wide cDNA microarray analysis to search for potential regulatory genes involved in cancer invasion and metastasis [53]. CNTN1 was discovered as an immediate downstream effector of the vascular endothelial growth factor (VEGF)C-VEFGR3/Flt4 that induced invasion and metastasis in lung cancer via the Src/p38 MAPK-mediated C/EBP signaling pathway (Figure 4). The activation of VEGF-C/Flt4 axis robustly upregulated CNTN1, leading to rearrangement of F-actin containing microfilaments bundles important in cell motility, enhanced cell invasion in vitro, and higher frequency of lung metastasis formation in vivo. The expression of CNTN1 was significantly correlated with Flt expression, tumor stage, lymph node metastasis, and patient survival in lung adenocarcinoma patients. The vascular endothelial growth factor (VEGF)/VEGF-receptor signaling pathway is integrally involved with growth, invasion, and metastasis of carcinomas through promoting angiogenesis and lymphangiogenesis [55]. During metastasis into lymph node, newly formed lymphatic capillaries, or pre-existing afferent lymphatic vessels allow passage for dissemination of tumor cells. The VEGFC-VEFGR3/Flt4 is a lymphatic specific biochemical axis with extracellular VEGFC as ligands, cell membrane VEGFR3 (Flt4) as receptor, and extracellular or intracellular pathway-related molecules as executors. The VEGFR3 (Flt4) mediate dissemination of several tumors and was found to be overexpressed in prostate [74], lung [75], colorectal [76], and head and neck cancers [77]. As well, VEGFR3 (Flt4) expression was found to correlate with different stages of cervical carcinogenesis [78]. Accumulating evidence suggest that the up and downstream effectors of the VEGFC-VEFGR3/Flt4 axis form a complicated biochemical network which work in synergy to facilitate tumor lymphangiogenesis and lymphatic metastasis. In this regard, it is reasonable to speculate that CNTN1 may be involved as an executor to facilitate this process.

CNTN1′s actions in this aspect may be mediated via facilitation of epithelial-to-mesenchymal transition (EMT). Epithelial cells depend on a fine-tuned and tightly regulated EMT for conversion into mesenchymal state during development [79]. Cells undergoing EMT adopt properties such as enhanced migratory and invasive abilities, increased apoptosis resistance, and extended production of extracellular matrix elements, all of which promotes cancer metastasis. In tumor cells, local cancer epithelial cells take over the evolutionarily conserved EMT process to lose their polarity and cell-cell contact while undergoing a dramatic cytoskeletal remodeling to acquire an invasive, well-defined mesenchymal phenotype [80], including the common loss of E-cadherin and overexpression of mesenchymal proteins (N-cadherin, vimentin, and fibronectin) [79,81]. Yan et al. discovered that while the knockdown of CNTN1 did not influence lung cancer cell proliferation, it significantly reduced cancer cell’s invasive abilities both in vitro and in xenograft model [58]. Specifically, CNTN1 fulfils this metastasis-promoting role in part through inhibition of E-cadherin. Although this suppression is likely mediated by the upregulation of transcriptional factors SIP1 and Slug, there’s also evidence that CNTN1 may downregulate E-cadherin at the gene level by inhibition of the pleckstrin homology (PH) domain and leucine-rich repeat protein phosphatase 2 (PHLPP2) instead of phosphatase and tensin homolog (PTEN), in turn resulting in AKT activation (Figure 4) [58]. AKT activity is upregulated by both upstream and downstream phosphatases, PTEN and PHLPP [82]. The activation of AKT reduces E-cadherin and constitutes a major component of EMT in promoting cancer cell invasion and metastasis.

Intriguingly, CNTN1 is also important in promoting chemo-resistant properties in lung cancer cells via induction of EMT. BCT-100 is an anti-cancer agent effective in treating arginine auxotrophic tumors including small cell lung cancer (SCLC) [62]. In a gene chip assay analyzing BCT-100-resistant cells in comparison to parent cells, CNTN1 was robustly upregulated and induced an EMT phenotype in resistant cells via targeting of the AKT pathways. Further functional analysis demonstrated that CNTN1 silencing re-sensitized resistant cells to BCT-100 treatment as well as attenuated the EMT phenotypes in resistant cell lines [57]. Similarly, CNTN1 expression was elevated in multidrug resistance (MDR) A549/cisplatin (A549/DDP) cells compared to its progenitor A549 lung cancer cells, and the silencing of CNTN1 rendered both cells higher cisplatin sensitivity and upregulated cisplatin-induced apoptosis, leading to inhibition of tumor invasion and metastasis [59]. This enhanced chemoresistance of lung cancer cells induced by CNTN1 overexpression was also found to be attributable to PI3K/AKT activated EMT enhancement (Figure 4). Indeed, CNTN1 downregulation successfully hindered cisplatin resistance and malignant progression via partly inactivating the EMT process by regulating the PI3K/AKT pathway. This finding was corroborated clinically by the positive association between CNTN1 expression and lymphatic invasion in non-small-cell lung carcinoma (NSCLC) patients that have received adjuvant cisplatin- or carboplatin-based treatments following surgery [72]. The tobacco carcinogen 4-(methylnitrosamino)-1-(3-pyridyl)-1-butanone (NNK) was also found to upregulate the expression of CNTN1, leading to enhanced lung cancer cells invasion via the activation of α7 nicotinic acetylcholine receptor (α7 nAChR) downstream of the AKT and extracellular signal-regulated kinase (ERK) signaling pathway [56]. The involvement of CNTN1 in promoting EMT and AKT suggest a potential involvement in its facilitation of cancer stem cell-like characteristics such as migration and anoikis resistance in cancer cells.

### 3.2. Gastric and Upper Gastrointestinal Cancers

Following the discovery of CNTN1′s oncogenic role in lung cancer, it has become increasingly clear that its role in the induction of invasion and metastasis may be a shared commonality in other cancers as well. In vitro studies found that the knockdown of Flt4 in human MKN45 gastric cancer cells with short hairpin RNA directly downregulated CNTN1 expression [60], which validates the role of CNTN1 in the VEGFC-VEFGR3/Flt4 axis. Compared to adjacent normal gastric mucosal tissues, both CNTN1 mRNA and protein expression were significantly upregulated in primary lesions and its expression level was positively correlated with VEFGC and Flt4 expression as well as lymphatic invasion, lymph node metastasis, TNM (tumor, node, metastasis) stage and worse prognosis in gastric cancer patients [60]. The induction of CNTN1 by the VEGFC-VEFGR3/Flt4 signalling axis is a result of the VEGFC-facilitated binding of CCAAT enhancer binding protein α (CEBPA) with the CNTN1 promoter [53] (Figure 4). This association was also validated in esophageal squamous cell carcinoma (ESCC) [66]. Recent study has shown that a reduction in CNTN1 expression alone was sufficient to reverse the enhanced migration ability of esophageal cancer cells attributable to ectopic VEFGC expression in vitro. As well, ESCC tissues showed significant upregulation of CNTN1 expression compared to adjacent non-tumor tissues and this increased expression was positively correlated with ESCC stage, lymph node metastasis and lymphatic invasion, supporting CNTN1′s role in esophageal cancer progression [70]. In a study investigating oral squamous cell carcinoma (OSCC), Flt-4 stimulation or inhibition directly resulted in upregulated or downregulated expression of CNTN1 respectively [71]. In vitro cell proliferation and migration assays further demonstrated that this association was accompanied by enhanced or inhibited cell proliferation and migration. The VEGFC-mediated lymphangiogenesis is suggested to be maintained via a paracrine mechanism in which the Flt4 expressed on the lymphatic endothelial cells is activated via its association with tumor cell-derived VEGFC [54]. Shigetomi et al. demonstrated that CNTN1 expression was directly regulated by Flt4 stimulation or inhibition, and this alteration was robustly associated with enhanced or reduced OSCC cell proliferation and migration correspondingly [54]. This finding suggests that the VEGFC-VEFGR3/Flt4 axis in tumor microenvironment may promote cell proliferation and migration via upregulating both VEGFC and CNTN1 in an autocrine manner, in turn contributing to cancer progression through the development of lymphatic metastasis. This process likely works in conjunction with its primary paracrine mechanism responsible for lymphangiogenesis in the lymphatic endothelial cells. Similar to ESCC, the upregulation of CNTN1 expression in OSCC patients was associated with poor overall survival (OS) [71]. The OSCC cells with CNTN1 knocked-down exhibits significant reduction in invasion but not proliferative properties [71], which is in line with previous findings demonstrating that CNTN1 may promote progression of cancer cells through exclusive activation of metastatic pathways.

### 3.3. Prostate Cancer

Accumulating evidence supports a pivotal role of cancer stem cells (CSC) in tumor progression and metastasis [83,84,85]. Prostate cancer stem cells (PCSC) have been identified as a major player underlying castration-resistant prostate cancer (PCSC) and associate with bone metastasis and poor prognosis [86,87]. Yan et al. was the first to describe a contribution of CNTN1 in prostate cancer stem cell like (PCSC)-derived tumor initiation [64]. The expression of CNTN1 was upregulated in PCSCs compared to non-stem prostate cancer cells with a concurrent downregulation of E-cadherin. This may provide a novel perspective for CNTN1′s role in CSCs in facilitating the acquisition of a more invasive cell phenotype. These observations, on the other hand, shed light on a potential mechanism leading to CNTN1 expression in cancers, i.e., the plasticity of PCSCs is likely a contributor. Disruption of the epithelial homeostasis with EMT is a major venue contributing to prostate cancer cells becoming invasive and metastasize [81]. As well, ectopic CNTN1 overexpression promoted cellular invasion in vitro and enhanced xenograft formation in vivo. Similarly, CNTN1 overexpression in PC cells produced more and larger lung nodules in a lung metastasis model compared to empty vector (EV) controls. Analysis of gene expression of CNTN1-knockdown in PCSCs showed predominantly affected genes to function in tumorigenesis, suggesting potential CNTN1 modulation of the expressions of tumorigenesis-relevant genes. Clinically, while CNTN1 expression was negative or low in normal prostate glands, it was upregulated in advanced prostate carcinomas and are clearly present in both primary prostate tumors and matched lymph node and bone metastasis.

This association was corroborated by a recent study in which elevated CNTN1 expression was found to positively associate with larger tumor size, advanced tumour stage, higher risk of metastasis and shorter overall survival (OS) in PC patients following radical prostatectomy [63]. Further analysis showed that silencing or knockdown of CNTN1 expression resulted in suppression of proliferation, invasion and migration capabilities in PC cells concurrent with a reduced PI3K/AKT signaling. CNTN1 inhibition also increased expression of E-cadherin while reduced N-cadherin and Vimentin, suggesting an inhibition of the EMT process pivotal in cancer metastases (Figure 4). Furthermore, docetaxel (Dox)-resistant PC cells exhibited upregulated CNTN1 expression and an EMT phenotype compared to parental cells [73]. Silencing of CNTN1 with short hairpin RNA (shRNA) inhibited cellular malignant phenotype and reduced expression of pluripotent markers such as CD44, octamer-binding transcription factor 4 (OCT-4), and Nanog. CNTN1 silencing sensitized Dox-resistant PC cells and inhibited PI3K/AKT signaling in vivo. These observations are in accordance with elevated CNTN1 expression found in drug resistant SCLC cells compared with progenitor cells [57], indicating that CNTN1 may function to promote a more malignant and cancer stem cell-like characteristics with higher migration ability and potentially anoikis resistance in cancer cells.

### 3.4. Breast Cancer

In a study investigating breast cancer (BC), ectopic CNTN1 overexpression enhanced cell proliferation, invasion, migration as well as cell cycle progression from G1 to S phase in breast cancer cells in vitro [88]. Notably, CNTN1 overexpression also enhanced breast cancer xenograft tumor growth in vivo in nude mice. However, a potential relationship of CNTN1 with primary BC has not been thoroughly investigated. To address this issue, we have taken advantage of the availability of numerous BC genetic databases; one of them is bc-GenExMiner 4.5 [89,90,91], which contains 10,716 BC tissues with gene expression determined by DNA microarray and 4712 BC samples profiled for gene expression using RNA-sequencing (RNA-seq). The dataset contains a total of *n* = 15428 BCs from multiple independent studies (cohorts; *n* = 57, 54 DNA microarray and three RNA-seq studies); importantly, data has been filtered by experts in the field for cross cohort analyses.

Based on gene expression, breast cancers are classified as five intrinsic subtypes: luminal A and B (ER+), human epidermal growth factor receptor 2 (HER2)-enriched (HER2-E), normal-like, basal-like, and claudin-low (the latter two being triple negative) [92,93,94,95]. Both HER2-E and basal-like tumors are aggressive subtypes [96]. Among the 5 subtypes, we noticed a significant higher level of CNTN1 in HER2-E in a population consisting of 50 DNA microarray studies (*n* = 9254) and in the population of three RNA-seq investigation (*n* = 4712) (Figure 5). The levels of CNTN1 expression vary among BC subtypes with different aggressiveness. For instance, Luminal B is a more aggressive BC sub-type compared to Luminal A but with reduced CNTN1 expression (Figure 5). However, HER2-E BCs are more aggressive compared to Luminal and Normal-like BCs [96] and express a significant higher level of CNTN1 than these sub-types of BCs (Figure 5). While these analyses reveal factors in addition to CNTN1 contributing to BC aggressiveness, evidence (Figure 5) supports an association of CNTN1 with this BC characteristics. Using both populations (DNA microarray and RNA-seq), we have analyzed CNTN1-correlated gene expression with correlations (*r* ≥ 0.4) determined by Pearson correlation and performed Gene Ontology (GO) enrichment analysis. In both positively correlated gene populations, enrichment in the terms of extracellular matrix organization (GO:0030198) and cell adhesion (GO:0007155) in Biological Process (BP) was observed in BC (All patients, Table 2). We further analyzed GO enrichment based on the intrinsic subtypes of BC and found robust enrichments in both GO terms (GO:0030198 and GO:0007155) in all BC intrinsic subtypes (Table 2). These analyses provide appealing evidence supporting the oncogenic function of CNTN1 in promoting BC via enhancing BC cell adhesion and interaction with extracellular matrix. Adhesion to bone marrow stromal components are critical for homing and surviving in bone marrow and thus metastatic outgrowth of bone metastasis [97,98]. Bone is a common site of BC metastasis [97,98]; in this regard, the enrichment of GO:0001501/skeletal system development in genes that are positively associated with CNTN1 expression in BC (Table 2) supports CNTN1′s role in facilitating BC bone metastasis. This possibility is supported by CNTN1-mediated promotion of prostate cancer metastasis [64] as prostate cancer predominantly metastasizes to the bone [99]. Collectively, our analyses suggest important functions of CNTN1 in promotion of BC progression and metastasis in part through enhancing BC cell adhesion.

### 3.5. Other Cancers

In hepatocellular carcinoma (HCC), CNTN1 expression was markedly increased in HCC tumors and this upregulation was correlated with tumor size and TNM stage [68]. As well, CNTN1 expression was found to be an independent prognostic factor for OS (hazard ratio 2.383, 95% confidence interval 1.262–4.503; *p* = 0.007) and DFS (disease free survival) (hazard ratio 2.356, 95% confidence interval 1.370–4.049; *p* = 0.002) in HCC patients. RET/PTC3 (Ret proto-oncogene and Ret-activating protein ELE1) rearrangement is the most frequent genetic alteration in thyroid cancer and most functions of RET are mediated through pathways including ERK, JNK, and PI3K/AKT [100,101,102,103]. CNTN1 was discovered as a potential RET-PTC3 downstream gene in GEO profile database and RET inhibitor regorafenib successfully downregulated CNTN1 expression in thyroid cancer cells [7]. In a study exploring the global gene and miRNA expression profiles of neuroblastoma cells after bortezomib treatment, CNTN1 was found to be downregulated along with RET, further confirming this relationship [104]. Interestingly, in a study exploring the therapeutic efficacy of *Ganoderma lucidum* extract (GLE), a potentially effective tumor growth inhibitor, CNTN1 was identified to be a part of the hub mRNA down-expressed in GLE-treated Hepa1–6-bearing C57BL/6 mice, mediated by miRNA mmu-miR23a-5p [105].

CNTN1 expression was found to be elevated in thyroid tumor tissues in comparison to its pre-cancer counterparts, and this alteration was particularly evident in the junctional zones where normal glandular structure and tumorous structure meets. This upregulation of CNTN1 was positively correlated with tumor size, TNM and tumor stage. Functionally, CNTN1 silencing in vitro restrained thyroid cancer cell migration and invasion abilities.

In a similar manner, significant correlation of CNTN1 expression with tumor size and TNM stage was also observed in stomach cancer [65]. While normal astrocytes did not express CNTN1, glioblastomas showed an overexpression of CNTN1 protein and a positive association was identified between increasing malignancy grade and CNTN1 expression [69]. Recent evidence identified CNTN1 as a novel melanoma-associate biomarker [106]. This is consistent with CNTN1′s role as an activator of Notch signaling and the association of Notch1 activation with a more metastatic phenotype in primary melanoma cells (Figure 4) [107]. The Notch signaling pathway is highly conserved, and its dysregulation is associated with many cancers [108]. Accumulating evidence have suggested Notch to be a key regulator in cancer stem cells (CSC) [109]. CNTN1 was identified as a ligand for NOTCH1 and activates Notch signaling by the release of notch intracellular domain (NCID) [14]. The knockdown of CNTN1 suppresses cellular proliferation and invasion in thyroid cancer and also inhibits the expression of cyclin D1 (CCDN1) [7], a Notch target gene, suggesting CNTN1 may participates in the tumor proliferation and metastasis in part via regulating the Notch pathway.

By taking advantage of the recently established GEPIA2 program with RNA sequencing data from The Cancer Genome Atlas (TCGA) and Genotype-Tissue Expression (GTEx) projects, we performed survival analyses with respect to CNTN1 expression across all 33 TCGA cancer types [110]. Our initial screen revealed a significant association between elevated CNTN1 and reduced overall survival (OS) in all cancer types (Figure 6a). With that, we performed detailed OS analysis for all cancer types, significant association between elevated CNTN1 with reduced survival was identified in bladder urothelial carcinoma (BLCA) (Figure 6b), brain lower grade glioma (LGG) (Figure 6c) and stomach adenocarcinoma (STAD) (Figure 6d). While these analyses support a positive association between high CNTN1 expression and poor OS in cancers, this association might be cancer-type specific at least at the level of mRNA; high CNTN1 expression displays a reverse association with poor OS in LGG (Figure 6c).

Mounting evidence in the past decade have supported CNTN1′s unique role in promoting tumor progression and metastasis in cancers (Table 1). Although detailed molecular mechanisms underpinning CNTN1′s oncogenic role remains incompletely understood, it is likely CNTN1 facilitate cancer migration and invasion through a combination of these aforementioned mechanisms (Figure 4).

## 4. Perspectives

In the last decade, CNTN1 has emerged as a promising oncogene in multiple cancer types. However, its oncogenic potential has not been thoroughly investigated, which limits our recognition of CNTN1′s utility in cancer diagnosis and therapy. In support of its therapeutic potential, a set of xenograft models reinforces CNTN1′s functionality in promoting tumorigenesis. To further investigate the contributions of CNTN1 in tumorigenesis and progression, a more physiologically relevant model that recapitulate the heterogeneity and tumour microenvironment would be a good step forward to advance current research. Considering CNTN1′s role in facilitating cell invasion and metastasis, it may be of interest to generate CNTN1-relevant conditional knock-in transgenic models to further explore the causal link between CNTN1 with cancer progression and metastasis. For example, a Cre-LoxP system can be explored in which mice carrying the transgene insertion of a strong translational and transcriptional termination (STOP) sequence flanked by loxP or FRT sites between the promoter sequence and CNTN1 which can be crossed with mice carrying tissue-specific promoters [111]. When Cre or FLP recombinase are expressed and present, the STOP cassette will be removed, allowing CNTN1 expression in the specific tissue of interest. This precise mediation of gene expression may allow a more informative investigation in the functionality of CNTN1 in carcinomas.

The involvement of CNTN1 in cancer formation and progression is supported by its common amplification in cancers and the association of this upregulation with the clinical features of cancers. The expression level of CNTN1 showed a clear association with clinicopathological parameters and poor prognosis in many cancers including lung [58], gastric [60], prostate [63], astrocytic gliomas [69], stomach [65], thyroid [7] cancers, hepatocellular carcinoma (HCC) [68] and oral squamous cell carcinoma (OSCC) [54]. As well, CNTN1 was discovered to be a novel biomarker for melanoma [106], glioblastoma [112], and colorectal cancer [113]. High CNTN1 expression was associated with biochemical recurrence following radical proctectomy in PC [64]. However, a competing endogenous RNA network study reported downregulation of CNTN1 in metastatic PC compared to primary PC [114]. In a high-throughput analysis from The Cancer Genome Atlas (TCGA) database, CNTN1 was identified as one of six pivotal prognostic EMT-related genes (ERGs) to affiliate with pro-oncogenic pathways and show a significant prognostic value in gastric cancer through OS analysis [61]. Similarly, an in-silico analysis of differentially expressed genes with functional protein products of NSCLC in five independent databases identified CNTN1 among with six other protein to associate with shorter OS and recurrence-free survival for predicted high-risk groups of NSCLC [115].

The upregulation of CNTN1, its association with worse clinical features, and its functionality in promoting cancer progression suggest CNTN1 being a potential target of cancer therapy. CNTN1′s physiological role as a neuronal cell adhesion molecule was likely explored by tumorigenesis. In view of the critical aspect of communications between cancer and stroma, proper adhesion is critical for cancer evolution or progression [97,98], targeting CNTN1 might thus be a useful option. However, it is important to note that while several drugs targeting cell adhesion molecules (e.g., αV- or β1-integrins) have shown promising outcomes in suppressing tumor growth in preclinical models, many of them have not significantly improved the disease-free or OS in patients [116]. Nonetheless, some encouraging findings have been made. For example, the anti-α_v_ antibody Intetumumab showed a non-significant trend towards improvements in OS compared to dacarbazine alone in a randomized, phase II study for patients with stage IV melanoma [117]. As well, the pan- αV inhibitor abituzumab reduced the cumulative incidence of bone lesion progression compared to placebo arm in a randomized phase II trial for patients with metastatic castration-resistant PC, despite not significantly extending progression-free survival (PFS) [118]. Nonetheless, these clinical trials illustrate that directly targeting cell adhesion molecules alone may not be sufficient to bring survival advantages to patients, suggesting a need for combinational therapies involving cell adhesion-based treatment.

A potential combination is with the emerging immunotherapy. In this regard, studies have investigated the potential of using immunotherapies in conjunction with integrin inhibitors to improve therapeutic responses. Preclinical mouse models have showed that combined administration of a fusion protein, consisting of a Fc domain and integrin targeting peptide, with albumin/IL-2 or anti-PD-1 immunotherapy was able to significantly enhance anti-tumor immunity and improve survival [119]. CNTN1 is an attractive target of cancer therapy. CNTN1 physiological expression in the central nerve system and its upregulation in multiple cancer types may qualify it as a tumor-associated antigen (TAA). TAAs are well-regarded targets for developing immunotherapy [120]. CNTN1 was identified as a TAA along with other four TAAs (CRKII, CFL1, NME2, and TKT) in IDTH1-mutant lower grade gliomas through a proteomic and immunologic approach [121]. CNTN1 triggered the activation of endogenous T cells in a substantial proportion for up to 56% of patients with astrocytic and/or oligodendroglial IDH1-mutant lower grade gliomas, but none in healthy donors. Furthermore, the extracellular exposure of CNTN1 is an appealing feature for developing immunotherapies. It is thus intriguing to envisage the possibility of immunotargeting of CNTN1 with a concurrent inhibition of cancer cell adhesion.

Likely, mechanisms leading to CNTN1 upregulation in cancers are complex. For instance, while the VEGFC/Flt4 pathway unregulated CNTN1 expression in lung cancer [53], CNTN1 upregulation in prostate cancer is an outcome of the conversion of non-CSC PC cells to PCSC and CNTN1 in turn contributes to PCSC-derived metastasis [64,122]. The connection of CNTN1 to PCSC is appealing. CSCs play a crucial role in cancer recurrence, metastasis and acquisition of resistance for chemotherapy and radiotherapy [83]. The maintenance of CSCs in its undifferentiated stage largely relies on the stem cell niche of structural moieties such as E-cadherin and stemness promoting pathways including Wnt Notch, transforming growth factor (TGF)-β, and nuclear factor (NF)-Kb [123,124,125]. Collectively, future research should focus on the connections of CNTN1 with CSCs. While detailed mechanisms governing CNTN’s oncogenic functions require further investigations, its physiological role in neuron development via CNTN’s adhesion properties suggests this theme being possibly hijacked by tumorigenesis. This potential is in agreement with the robust enrichment of genes that are positively correlated with CNTN1 expression in the GO terms of “extracellular matrix organization” and “cell adhesion” (Table 2). The potential mechanisms underlying this adhesion modification remains unknown. Those of interaction proteins in neuron development (Figure 3) may shed light on the mechanisms responsible CNTN1-derived modulation of cancer adhesion. As CNTN1 lacks of a intracellular region (Figure 2 and Figure 4), it remains a possibility that CNTN1′s interaction with proteins functioning in tumorigenesis may include those residing in the lipid bilayer of cell membrane. Regardless the nature of CNTN1 binding proteins, identification of these is an appealing avenue of future investigation.

## 5. Conclusions

In recent decades, drastic improvement in strategies for detection and treatment of cancer patients have contributed to a marked decrease of cancer mortality worldwide. In the concept of “molecular targeted therapy”, correcting for alterations that occur in cancer progression and metastasis remains the fundamental goal. Changes in the intercellular adhesive properties of cancer cells as a consequence of altered CNTN1 expression certainly play a role in tumorigenesis. Positioned at the crossroad of pathways associated with cancer invasion and metastasis, CNTN1 represents a promising target for cancer therapy. Indeed, silencing or inhibition of CNTN1 in experimental models have already highlighted its suitability as a potential target in lung, gastric, prostate, thyroid cancers, and oral squamous cell carcinoma. Like all successful molecular targeted therapy, an effective CNTN1-targeted therapy in cancer patients would benefit from a rational selection of delivery contexts. Although CNTN1 only show limited expression in normal tissues, it is hard to know whether targeting CNTN1 in a tissue specific manner may have adverse clinical outcomes.

## Figures and Tables

**Figure 1 genes-11-00874-f001:**
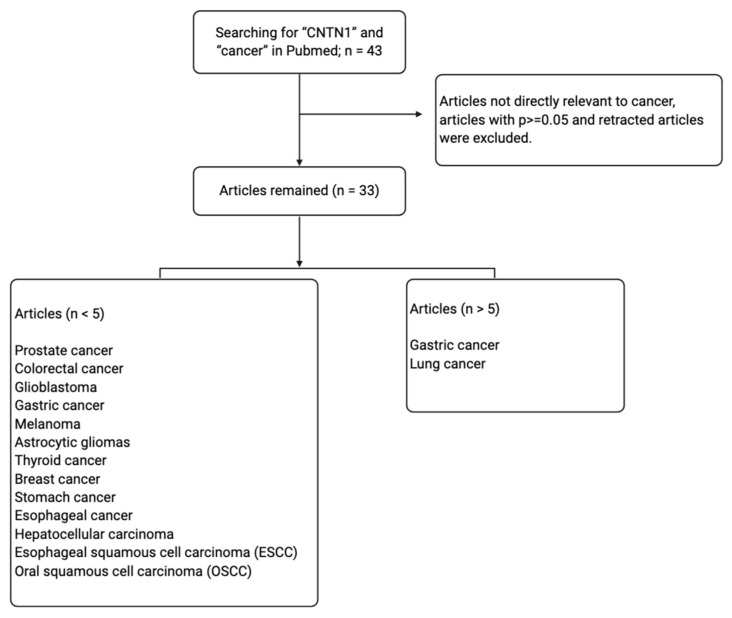
Systemic literature searching conditions and selection of articles for review.

**Figure 2 genes-11-00874-f002:**
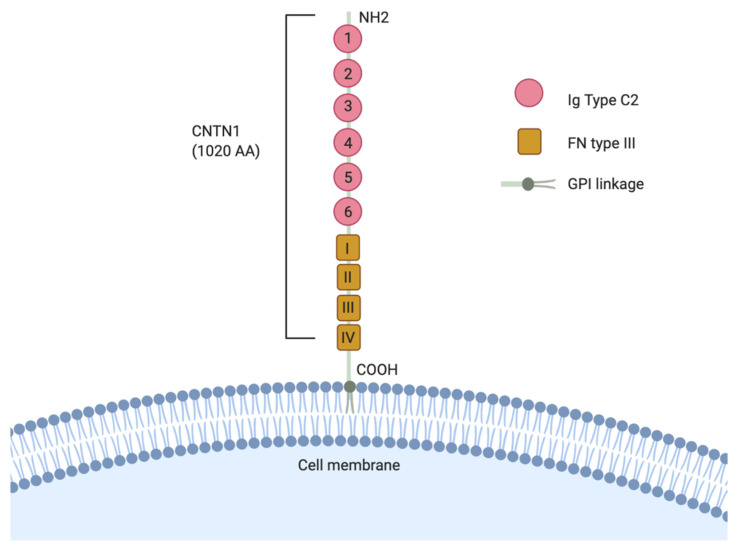
Schematic representation of the Contactin 1 (CNTN1) structure. CNTN1 is a 1020 amino acids (AA) protein comprised of six immunoglobulin (Ig)-like repeats followed by four fibronectin (FN) type III-like domains and is connected by a glycosylphosphatidylinositol (GPI)-anchor to the plasma membrane.

**Figure 3 genes-11-00874-f003:**
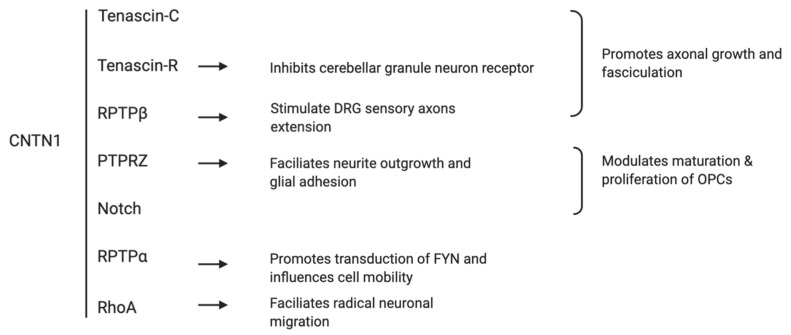
Schematic summary of traditional extracellular interactions of CNTN1.

**Figure 4 genes-11-00874-f004:**
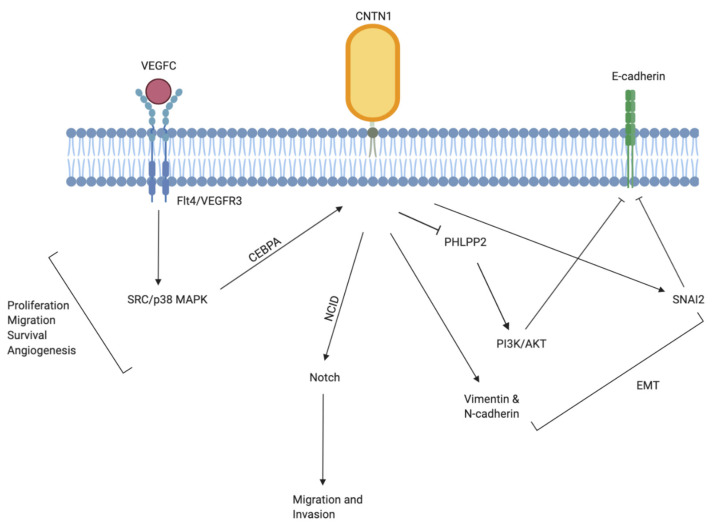
Graphic illustration of the currently known mechanisms underlying CNTN1-induced migration and invasion of cancer cells. CNTN1 acts as a downstream effector of the VEGFC-VEFGR3/Flt4 axis, leading to further activation of the CCAAT enhancer binding protein α (CEBPA) facilitated SRC/p38 mitogen-activated protein kinase (MAPK) signaling. CNTN1 activates Notch signaling via interacting with NOTCH1 and notch intracellular domain (NCID). CNTN1 inhibits E-cadherin in part via transcriptional regulation of SIP1 and Slug as well as AKT activation. The inhibition of E-cadherin and upregulation of Vimentin and N-cadherin are characteristic events of EMT.

**Figure 5 genes-11-00874-f005:**
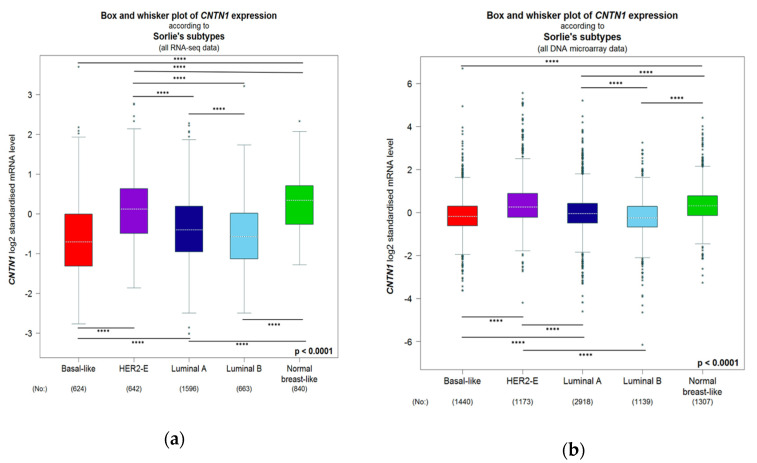
Box and whisker plot of CNTN1 expression examined by bc-GenExMiner 4.5 [91] from RNA-seq (**A**) and DNA microarray (**B**) separated by breast cancer subtypes. Global significant difference between groups was assessed by Welch’s test, followed by Dunnett–Tukey–Kramer’s test for pairwise comparison; **** *p* > 0.0001.

**Figure 6 genes-11-00874-f006:**
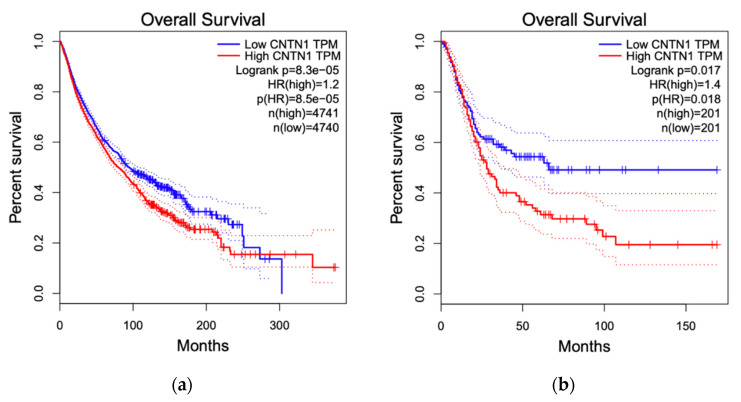
CNTN1 associates with alterations in overall survival (OS). Survival analyzes were performed on all 33 The Cancer Genome Atlas (TCGA) cancer types organized in the Gene Expression Profiling Interactive Analysis (GEPIA2) website. OS with respect to CNTN1 expression was shown for all 33 TCGA cancer types (**a**), BLCA (**b**), LGG (**c**), and STAD (**d**). BLCA: bladder urothelial carcinoma; LGG: brain lower grade glioma; STAD: stomach adenocarcinoma. Group cut-off was set at 50% and hazard ratio was calculated based on Cox Proportional-Hazards Model.

**Table 1 genes-11-00874-t001:** Changes in CNTN1 expression in cancer and correlation with tumor progression.

Tumor Type	CNTN1 Expression in Cancer and Major Findings	References
Lung cancer	Knockdown of CNTN1 reduced cell invasion but not proliferation. CNTN1 downregulates E-cadherin via protein kinase B (AKT) activation in part by preventing PHLPP2-mediated AKT dephosphorylation.	[58]
	CNTN1 promotes cisplatin resistance in cisplatin-resistant lung cancer cells through activation of the phosphatidylinositol 3-kinase(PI3K)/AKT signalling pathway. CNTN1-knockdown in resistant cells partly reversed the epithelial-mesenchymal transition (EMT) phenotype, increased drug sensitivity, and attenuated malignant progression.	[59]
	NNK upregulated the expression of CNTN1 and increased invasive abilities of lung cancer cells by α7 nicotinic acetylcholine receptor (α7 nAChR) and extracellular signal-regulated kinase (ERK) signalling pathway.	[56]
	CNTN1 was upregulated in BCT-100-resistant cell lines and promoted EMT progression via AKT pathway. Silencing of CNTN1 re-sensitized resistant cells to BCT-100 treatment and attenuated the EMT phenotype.	[57]
	CNTN1 is upregulated by vascular endothelial growth factor C (VEGFC)-VEGF receptor 3 (VEFGR3)/fms-related tyrosine kinase 4 (Flt4) axis activation via the Src/p38 mitogen activated protein kinase pathway (MAPK)-mediated CCAAT enhancer binding protein (C/EBP) signalling pathway. Expression of CNTN1 correlated with tumor stage (*p* = 0.048), lymph node metastasis (*p* < 0.001), and patient survival (*p* < 0.001) in lung adenocarcinoma. Knockdown of CNTN1 reduced cell invasion capacity in vitro and reversed the enhanced lung metastasis attributable to VEGF-C overexpression in vivo.	[53]
	CNTN1 expression was upregulated in multidrug resistance (MDR) A549/cisplatin (C549/DDP) cells compared to parental A549 cells. Silencing of CNTN1 rendered cells more sensitive to cisplatin and increased cisplatin-induced apoptosis; CNTN1 knockdown reduced metastasis and invasion but not cell proliferation. In non-small-cell lung carcinoma (NSCLC) patients (*n* = 143) that have received adjuvant cisplatin-or carboplatin-based treatment after surgery, CNTN1 expression positively correlated with lymphatic invasion.	[72]
Gastric adenocarcinoma (GAC)	CNTN1 was identified as an EMT-related gene (ERG) and showed significant prognostic value in GAC (*p* value = 0.006; Hazard ratio = 1.397 (1.098-1.776). Upregulation of CNTN1 is associated with worse pathologic grade (*p* = 0.049) and negatively correlate with survival time of GAC patients (Log-rank *p* = 0.011, hazard ratio = 1.5)	[61]
	Gastric tumors showed higher level of CNTN1 mRNA compared to noncancerous gastric samples (*p* = 0.01) and this upregulation was correlated with VEGFC (*p* < 0.001), VEGFR3 (*p* = 0.001). Tumor size (*p* = 0.035), TNM (tumor, node, metastasis) stage (*p* = 0.02), lymphatic invasion (*p* = 0.01), and lymph node metastasis (*p* < 0.01). Patients with CNTN1-positive tumors had a significant shorter survival time than those with negative tumors (log rank: *p* = 0.012) and higher lymphatic vessel density (LVD) (*p* < 0.001).	[60]
Oesophageal squamous cell carcinoma (ESCC)	CNTN-1 expression was significantly increased in the tumour tissue compared with the normal oesophageal tissue (*p* = 0.001) and was correlated with oesophageal squamous cell carcinoma stage (*p* = 0.006), lymph node metastasis (*p* = 0.018) and lymphatic invasion (*p* = 0.035)	[70]
	Stimulation of VEGFC induced cell growth and migration through CNTN1 in vitro. CNTN1 expression closely correlate with VEGFC in both ESCC tissues and normal esophageal tissues.	[66]
Oral squamous cell carcinoma (OSCC)	Flt-4 stimulation upregulated the expression of CNTN1 and VEGF-C in OSCC cells CNTN1 expression significantly correlated with VEGF-C and Flt-4, as well as with neck metastasis (*p* = 0.022), vascular invasion (*p* = 0.016) and lymphatic invasion (*p* = 0.022) in patients with tongue squamous cell carcinoma (*n* = 55).	[54]
	CNTN1 expression was significantly associated with regional lymph node metastasis of (*p* = 0.006), overall survival (*p* = 0.032; log rank test) and disease-free survival of OSCC patients (*p* = 0.038; log-rank test). CNTN1 ablation notably suppressed the invasion potential of OSCC cell lines.	[71]
Hepatocellular carcinoma (HCC)	Higher mRNA and protein CNTN1 expression was observed in HCC compared to adjacent tissues (*p* = 0.01) and associated with tumor size, tumor capsule, status of metastasis, and tumor–node–metastasis stage. High CNTN1 was correlated with reduced overall survival (OS) rate (*p* = 0.001) and disease-free survival (DFS) rate (*p* = 0.001).	[68]
Astrocytic Gliomas	CNTN1 were overexpressed in glioblastomas compared to normal brain and was associated with increased malignancy grade. Overexpression of CNTN1 in glioblastoma cells did not alter the proliferation rate.	[69]
Breast cancer	CNTN1 overexpression enhanced breast cancer cell proliferation, migration, invasion, cell cycle progression in vitro, and promoted xenograft tumor growth in vivo.	[62]
Thyroid cancer	CNTN1 expression is elevated in thyroid cancer tissues compared to paracancer tissues (*n* = 100) (*p* < 0.001) and associated with larger tumor size (*p* = 0.003), and TNM stage (*p* = 033). Knockdown of CNTN1 significant inhibited tumor proliferation and invasion.	[7]
Stomach cancer	CNTN1 expression significantly correlated with tumor size (*p* = 0.0003) and TNM (*p* = 0.0419).	[65]
Prostate cancer (PC)	CNTN1 knockdown reduced prostate cancer stem cells (PCSC)-mediated tumor initiation; CNTN1 overexpression enhanced PC invasion in vitro and promoted xenograft tumor formation and lung metastasis in vivo. CNTN1 upregulation led to elevated AKT activation and reduced E-cadherin expression. CNTN1-positive tumours (*n* = 637) showed advanced progression and worse biochemical recurrence-free survival following radical prostatectomy (*p* < 0.05).	[64]
	CNTN1 expression was upregulated in PC compared to adjacent tissues and was positively correlated with tumor size (*p* = 0.015), stage (*p* = 0.004), lymph node metastasis (*p* = 0.022), distant metastasis (*p* = 0.006) and a poorer prognosis in PC patients. CNTN1 knockdown resulted in significant inhibition of proliferation, colony formation, migration and invasion of PC cells in vitro. Silencing of CNTN1 suppressed EMT in PC cells via upregulation of E-cadherin, downregulation of N-cadherin and vimentin expression, concurrent with reduced PI3K/AKT activity.	[63]
	CNTN1 knockdown attenuated cell proliferation, migration, invasion and PI3K/AKT mediated EMT in docetaxel-resistant PC cells. Silencing of CNTN1 sensitized docetaxel-resistant PC cells in xenograft model, with concurrent inhibition of PI3K signalling and downregulated EMT markers.	[73]

**Table 2 genes-11-00874-t002:** GO enrichment of CNTN1-positively correlated genes.

Go ID	Term	*p*-Value	%Target List	Associated Genes
*All patients*				
GO:0030198	extracellular matrix organization	1.45×10 ^−5^	14.63	CCDC80, COL8A2, ECM2, FBN1, ITGA11, LOX
GO:0007155	cell adhesion	8.05×10 ^−5^	17.07	CCN4, CDH11, CNTN1, COL12A1, ITGA11, OMD, SPON1
*Basal-like*				
GO:0030198	extracellular matrix organization	2.61×10 ^−15^	14.08	ADAM12, APBB2, CCDC80, COL3A1, COL6A3, CRISPLD2, DCN, ECM2, FBLN1, FBN1, JAM2, JAM3, LAMA2, MATN3, MMP16, MMP2, NDNF, POSTN, SMOC2, VCAN
GO:0001501	skeletal system development	1.30×10 ^−8^	7.75	CDH11, CHRD, COL3A1, EVC, EBN1, IGF2, MATN3, MMP16, PTH1R, TLL1, VCAN
GO:0007155	cell adhesion	3.07×10 ^−8^	12.68	ADAM12, CDH11, CNTN1, COL6A3, EDIL3, FAP, FEZ1, JCAD, LAMA2, NUAK1, OMD, PCDH7, POSTN, SEMA5A, SPOCK1, SVEP1, THBS2, VCAN
*Her2-E*				
GO:0030198	extracellular matrix organization	3.45×10 ^−19^	21.74	ADAM12, COL1A1, COL1A2, COL3A1, COL5A1, COL5A2, COL8A1, COL8A2, CRISPLD2, DCN, DDR2, ECM2, FBN1, JAM3, LOX, LUM, MMP2, RECK, SPARC, VCAN
GO:0001501	skeletal system development	5.52×10 ^−7^	8.7	CDH11, COL1A2, COL1A2, COL3A1, COL5A2, EVC, FBN1, VCAN
GO:0007155	cell adhesion	1.05×10 ^−7^	15.22	ADAM12, CCN4, CDH11, CNTN1, COL5A1, COL8A1, DDR2, EDIL3, FAP, OMD, PCDH7, SPON1, THBS2, VCAN
*Luminal A*				
GO:0030198	extracellular matrix organization	2.84×10 ^−38^	9.83	ADAM12, ADAM19, BGN, CCDC80, COL10A1, COL11A1, COL14A1, COL16A1, COL1A1, COL1A2, COL3A1, COL5A1, COL5A2, COL5A3, COL6A1, COL6A2, COL6A3, COL7A1, COL8A1, COL8A2, COMP, CRISPLD2, CDN, DDR2, ECM2, EGFL6, ELN, FBLN1, FBLN2, FBLN5, FBN1, FN1, FOXF2, ITGA11, ITGB1, JAM2, JAM3, LAMA1, LAMA2, LAMA4, LAMB1, LAMC1, LOX, LOXL1, LUM, MATN3, MFAP2, MFAP5, MMP11, MMP13MMP14, MMP16, MMP19, MMP2, MMP27, NID1, NID2, PDGFRA, POSTN, PXDN, RECK, SH3PXD2A, SPARC, VCAN
GO:0001501	skeletal system development	1.25×10 ^−14^	11.67	ALX4, ARSE, BMP1, BMP7, CDH11, COL10A1, COL1A1, COL1A2, COL3A1, COL6A2, COMP, EVC, FBN1, FRZB, GJA5, GLI2, HOXA11, IGF1, IGF2, MATN3, MMP14, MMP16, NKX3–2, PTH1R, RASSF2, SH3PXD2B, SHOX2, TLL1, VCAN
GO:0007155	cell adhesion	6.51×10 ^−29^	3.53	ADAM12, ADAM23, BOC, CCN2, CCN4, CDH11, CERCAM, CNTN1, COL12A1, COL16A1, COL5A1, COL5A3, COL6A1, COL6A2, COL6A3, COL6A6, COL7A1, COL8A1, CXCL12, CYP1B1, DDR2, DPP4, DPT, EGFL6, EMILIN1, ENTPD1, FAP, FAT1, FEZ1, FN1, GAS6, GPNMB, ISLRITGA11, ITGB1, ITGBL1, JCAD, LAMA1, LAMA2, LAMA4, LAMB1, LAMC1, LOXL2, LSAMP, MFAP4, MFGE8, MXRA8, NID2, NRP2, NTM, NUAK1, OMD, PCDH18, PCDH7, PCDHB12, PCDHB18P, PCDHGA12, PCDHGC3, PLXNC1, POSTN, PPFIBP1, PRKCA, PTK7, ROBO1, SEMA5A, SPOCK1, SPON1, SPON2, SRPX, SSPN, TGFB1I1, THBS2, TLN2, TNFAIP6, VCAN, VCL
*Luminal B*				
GO:0030198	extracellular matrix organization	4.60×10 ^−38^	11.07	ADAM12, BGN, CCDC80, COL10A1, COL11A1, COL14A1, COL16A1, COL1A1, COL1A2, COL3A1, COL5A1, COL5A2, COL5A3, COL6A1, COL6A2, COL6A3, COL7A1, COL8A1, COL8A2, COMP, CRISPLD2, DCN, DDR2, ECM2, ELN, FBLN1, FBLN2, FBLN5, FBN1, FN1, FOXF2, HSPG2, ITGA11, JAM3, LAMA1, LAMA2, LAMA4, LAMB1, LAMC1, LOX, LOXL1, LUM, MFAP2, MFAP2, MFAP5, MMP11, MMP13, MMP14, MMP19, MMP2, NDNF, NID1, NID2, PDGFRA, POSTN, PXDN, RECK, SH3PXD2A, SPARC, VCAN
GO:0001501	skeletal system development	6.72×10 ^−17^	5.44	ARSE, BMP1, CDH11, COL10A1, COL1A1, COL1A2, COL3A1, COL5A2, COMP, DLX5, EN1, EVC, FBN1, FRZB, GJA5, GLI2, HOXA10, HOXA11, HOXA4, IGF2, MMP14, NKX3–2, PAX1, PRELP, PTH1R, SH3PXD2B, SHOX2, TLL1, VCAN
GO:0007155	cell adhesion	4.93×10 ^−25^	12.01	ADAM12, CCN4, CDH11, CNTN1, COL5A1, COL8A1, DDR2, EDIL3, FAP, OMD, PCDH7, SPON1, THBS2, VCAN
*Normal-like*				
GO:0030198	extracellular matrix organization	1.44×10 ^−5^	11.48	COL14A1, COL8A2, DDR2, JAM3, LAMA2, MMP16, RECK
GO:0007155	cell adhesion	9.99×10 ^−4^	11.48	CNTN1, DDR2, LAMA2, PCDH7, SEMA5A, SPON1, SSPN

Results were generated using the breast cancer genetic data and tools provided by bc-GenExMiner 4.5. The RNA-seq data was used. All patients: all breast cancers are included. Basal-like, Her2-E, Luminal A, Luminal B, and Normal-like are intrinsic subtypes of breast cancer.

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
