# Peer review of "Contactin 1: An Important and Emerging Oncogenic Protein Promoting Cancer Progression and Metastasis"

_genes, 2020, doi:10.3390/genes11080874_

Round 1
Reviewer 1 Report
This is a comprehensive review of the tumor-promoting role of contactin 1 (CNTN1) in many types of cancers by citing many pre-clinical and clinical reports, suggesting CNTN1 as a promising drug target.
- Survival data: The significance of CNTN1 can be supported by linking its expression level to overall survival. GEPIA is a newly developed interactive web server for analyzing the RNA sequencing expression data of 9,736 tumors and 8,587 normal samples from the TCGA and the GTEx projects, using a standard processing pipeline (ref).
(ref) Tang, Z. et al. (2017) GEPIA: a web server for cancer and normal gene expression profiling and interactive analyses. Nucleic Acids Res, 10.1093/nar/gkx247.
According to GEPIA survival data, a high CNTN1 transcript level has a significant impact (p=8X10^-5) for all patients (N=4741 for high expression, and N = 4740 for low expression). However, for breast cancer, prostate cancer, and many other cancers, a high CNTN1 expression level does not show a statistically significant difference. It is recommended to show this type of overall survival data to support the importance of CNTN1.
- Clinical usefulness: cell adhesion molecules have a varying role in the proliferation, migration, and invasion of tumor cells. However, none of the drug candidates, which are targeted to cell adhesion molecules, showed clinical usefulness in the phase III trial. It is recommended to evaluate the possibility of successful CNTN1-targeted therapy by comparing it with other clinical trials. If any combinatorial application is considered to increase clinical usefulness, it is recommended to describe it.
- Linkage to immunity: Since CNTN1 is a member of the immunoglobulin superfamily, it is recommended to describe any potential linkage of CNTN1 regulation to immunotherapy.
- Figure 5 shows the variations of CNTN1 expression levels in subtypes in breast cancer, but the interpretation is not given. Please explain why this figure is important.
Author Response
We thank reviewer #1 for the detail and insightful comments. Here are our point-by-point responses.
Response to comment #1 - We thank the reviewer for this recommendation. The survival data has been generated from GEPIA (reference 110). Significant associations were found for CNTN1 expression with respect to all patients, BLCA (bladder urothelial carcinoma), LGG (brain lower grade glioma) and STAD (stomach adenocarcinoma). This data is presented in Figure 6 (a new figure) and discussed (page 17, marked).
Response to comment #2 - We appreciate this insightful comment. While it is true that drug candidates targeting cell adhesion molecules have displayed a general lack of efficacy in clinical trials, some have shown marginal improvements in patient outcomes. For example, Intetumumab has shown a non-significant trend in improving OS in melanoma (ref 117) and abituzumab was observed to reduce the cumulative incidence of bone lesion progression in metastatic CRPC patients (ref 118). This knowledge derived from clinical trials is added (pages 19-20; lines 450 - 465).
Response to comment #3 - The aspect of targeting CNTN1 via immunotherapy has been discussed (page 20, lines 466-480). CNTN1 was recently identified as a TAA in IDTH1-mutant lower grade gliomas and triggered activation of endogenous T cells in a substantial proportion of LGG patients but not in healthy donors (ref 121). Interestingly, cell adhesion-based therapy in combination with immunotherapy has recently been reported, which showed promising results in enhancing anti-tumor immunity and improved survival in preclinical models (ref 119). This suggest that CNTN1 may be a useful target for immunotherapies and it may be worthwhile to consider a combinatorial application of CNTN1-targed therapy with immunotherapy.
Response to comment #4 - We thanks the reviewer for this comment. A brief discussion for this data is added on page 12 (lines 326-331).
Reviewer 2 Report
I appreciated the content of the article. Very well written and according to PRISMA guidelines. All the studies are referred to as with the appropriate reference number. Very well written. I would suggest only minor spell check.
Author Response
We appreciate the reviewer’s encouragement. The manuscript has been carefully edited by a native English
speaking colleague.
Reviewer 3 Report
Genes 2020-877670
Comments
This review article is focusing on Contactin1 (CNTN1) as a cancer biomarker.
In principle, the authors capture the importance of CNTN1 in promoting cancer
The exact mechanism of CNTN1 as a potential cancer biomarker in the literature is
still fragmented and often indirect.
The validation of CNTN1 as a target for drug discovery and development or as an immunotherapy target need to be rigorously addressed before, we will see actual effective cancer drugs and/or immunotherapeutic antibodies for cancers with upregulated CNTN1 expression,
However, the authors miss to include important recent articles covering this subject for example:
- Wang et al., 2020. Upregulation of contactin‑1 expression promotes prostate cancer progression.
- Chen et al., 2020. CNTN-1 promotes docetaxel resistance and epithelialto-mesenchymal transition via the PI3K/Akt signaling pathway in prostate cancer.
- Tang et al., 2020. Molecule mechanisms of Ganoderma lucidum treated hepatocellular carcinoma based on the transcriptional profiles and miRNA-target network.
- Luo et al., 2020. Prediction of potential prognostic biomarkers in metastatic prostate cancer based on a competing endogenous RNA regulatory
- Łuczkowska et al., 2020. Molecular Mechanisms of Bortezomib Action: Novel Evidence for the miRNA–mRNA Interaction Involvement.
- Dettling et al., 2018. Identification of CRKII, CFL1, CNTN1, NME2, and TKT as Novel and Frequent T-Cell Targets in Human IDH-Mutant Glioma.
It might be of interest to the readers to include and discuss these publications. Some of the tables/illustrations can be adjusted accordingly.
Author Response
We appreciate the reviewer’s comments. The missing references have been added and discussed.
Response to comment #1 - This article has been cited and discussed (ref 63; page 11, lines 295-297).
Response to comment #2 - The manuscript has been added and discussed (ref 73; page 11, lines 301-306).
Response to comment #3 - This article has been cited and discussed (ref 105; page 16, lines 367-369).
Response to comment #4 - This article has been cited and discussed (ref 114; page 18, lines 441-443).
Response to comment #5 - This article has been cited and discussed (ref 104; page 16, lines 369-372).
Response to comment #6 - This article has been cited and discussed (ref 121; page 19, lines 474-478).